# Characterization of an α-Amylase from the Honeybee Chalk Brood Pathogen *Ascosphaera apis*

**DOI:** 10.3390/jof9111082

**Published:** 2023-11-05

**Authors:** Jincheng Li, Sen Liu, Chenjie Yang, Nemat O. Keyhani, Huili Pu, Longbin Lin, Xiaoxia Li, Peisong Jia, Dongmei Wu, Jieming Pan, Philip C. Stevenson, G Mandela Fernández-Grandon, Liaoyuan Zhang, Yuxi Chen, Xiayu Guan, Junzhi Qiu

**Affiliations:** 1State Key Laboratory of Ecological Pest Control for Fujian and Taiwan Crops, College of Life Sciences, Fujian Agriculture and Forestry University, Fuzhou 350002, China; l1371891671@126.com (J.L.); m17633615410@163.com (S.L.); cjyang0525@126.com (C.Y.); hdpuhuili@163.com (H.P.); llb_77@126.com (L.L.); xiaoyaotan-lxx@163.com (X.L.); 000q121064@fafu.edu.cn (L.Z.); liesleyu@163.com (Y.C.); 2Department of Biological Sciences, University of Illinois, Chicago, IL 60607, USA; keyhani@uic.edu; 3Institute of Plant Protection, Xinjiang Academy of Agricultural Sciences, Urumqi 830091, China; jps-fly@163.com; 4Biotechnology Research Institute, Xinjiang Academy of Agricultural and Reclamation Sciences, Shihezi 832061, China; wdm0999123@sina.com; 5College of Biology & Pharmacy, Yulin Normal University, Yulin 537000, China; jiemingpan@163.com; 6Natural Resources Institute, University of Greenwich, Chatham Maritime ME4 4TB, UK; p.c.stevenson@greenwich.ac.uk (P.C.S.); m.fernandez-grandon@greenwich.ac.uk (G.M.F.-G.); 7College of Horticulture, Fujian Agriculture and Forestry University, Fuzhou 350002, China

**Keywords:** fungal pathogen, *Ascosphaera apis*, α-amylase, medium factor optimization, purification, large-scale production

## Abstract

The insect pathogenic fungus, *Ascosphaera apis*, is the causative agent of honeybee chalk brood disease. Amylases are secreted by many plant pathogenic fungi to access host nutrients through the metabolism of starch, and the identification of new amylases can have important biotechnological applications. Production of amylase by *A. apis* in submerged culture was optimized using the response surface method (RSM). Media composition was modeled using Box–Behnken design (BBD) at three levels of three variables, and the model was experimentally validated to predict amylase activity (*R*^2^ = 0.9528). Amylase activity was highest (45.28 ± 1.16 U/mL, mean ± SE) in media composed of 46 g/L maltose and1.51 g/L CaCl_2_ at a pH of 6.6, where total activity was ~11-fold greater as compared to standard basal media. The enzyme was purified to homogeneity with a 2.5% yield and 14-fold purification. The purified enzyme had a molecular weight of 75 kDa and was thermostable and active in a broad pH range (> 80% activity at a pH range of 7–10), with optimal activity at 55 °C and pH = 7.5. Kinetic analyses revealed a *K*_m_ of 6.22 mmol/L and a *V*_max_ of 4.21 μmol/mL·min using soluble starch as the substrate. Activity was significantly stimulated by Fe^2+^ and completely inhibited by Cu^2+^, Mn^2+^, and Ba^2+^ (10 mM). Ethanol and chloroform (10% *v*/*v*) also caused significant levels of inhibition. The purified amylase essentially exhibited activity only on hydrolyzed soluble starch, producing mainly glucose and maltose, indicating that it is an endo-amylase (α-amylase). Amylase activity peaked at 99.38 U/mL fermented in a 3.7 L-bioreactor (2.15-fold greater than what was observed in flask cultures). These data provide a strategy for optimizing the production of enzymes from fungi and provide insight into the α-amylase of *A. apis*.

## 1. Introduction

Chalkbrood is a fungal disease of bee brood resulting from infection by spores of the ascomycete *Ascosphaera apis* [1,2]. Unlike some broad host range entomopathogenic fungi (e.g., *Beauveria bassiana* and *Metarhizium anisopliae*) that germinate and penetrate the cuticles of their insect hosts [3,4], *A. apis* spores must be consumed to infect larvae [5]. All larval castes are susceptible to infection. However, adult bees are resistant, although they can act as vectors within and between hives. Once ingested, the spores germinate in the lumen of the gut, penetrate the gut wall, and grow inside the body cavity, extruding through the posterior end of the larva [6].

*Ascosphaera apis* produces a variety of hydrolytic enzymes, including proteases and glycosidases, to digest host proteins and carbohydrates, although for the most part, their functions and substrate specificities await characterization [7]. For example, several proteases and esterases have been identified in *A. apis*-infected larval hemolymph, although their exact role(s) in the infection process also remains unclear [6]. Starch is an essential storage compound in all organisms, can accumulate to high levels in specific tissues, and is a source of monosaccharides (energy) for *A. apis*. Intriguingly, honeybee nectar foragers are capable of utilizing starch as fuel for flight, whereas drones cannot [8,9]. Bee bread resulting from pollen conversion (similar to honey derived from nectar) is provisioned to honeybee larvae and contains proteins, sugars (including starch), and lipids [10,11].

Amylases are important degradative hydrolases involved in starch metabolism [12], although surprisingly little is known about their characteristics and production in *A. apis*. Amylases are also used in a variety of biotechnological applications, including starch saccharification, textile production and treatments, food, baking, brewing, pharmaceutics, detergents, and distillation industries [13,14,15,16,17]. Amylases are mainly classified into two major groups, α- and β-, according to the anomeric sugar product resulting from the enzymatic reaction [18,19]. α-Amylases are endo-enzymes that catalyze the hydrolysis of α-D-(1,4) glycosidic linkages in starch and related carbohydrates [16], whereas β-amylases are exo-enzymes that catalyze the hydrolysis of α-1,4-glucan bonds in amylosaccharide chains from the non-reducing ends of substrates, including starch, glycogen, amylose, amylopectin, and other malto-oligosaccharides to successively release β-anomeric maltose [20,21].

Here we report on the characterization of a secreted amylase from the fungal bee pathogen, *A. apis*. To better study the protein, a rational design strategy was used to optimize enzyme production from the fungus. Traditional production approaches targeting media composition involve a time-consuming and potentially expensive approach by which one factor is varied while other variables are maintained constant, whereas the method we used involves the application of the optimization algorithm [22], where imitations of single-factor optimization were reduced by employing response surface methodology(RSM) [23]. RSM merges experimental strategies with mathematical methods and statistical inference to determine and select specific experimental designs, build models, and explore relationships between response variable(s) and a set of design variables [22,23]. This method is increasingly being used to optimize various processes in food chemistry, material science, chemical engineering, enzyme applications, and microbiology [24,25,26]. The enzyme was purified from culture supernatants, and its activity was characterized.

## 2. Materials and Methods

### 2.1. Microorganism and Preparation of Inoculum

*Ascosphaera apis* used in the present study was isolated from honeybee larvae collected in an experimental beehouse at Fujian Agricultural and Forestry University. The strain was deposited in the China General Microbiological Culture Collection Center (CGMCC), China, with an accession number of CGMCC 3.17982. For preparation of inoculum, *A. apis* was initially grown on potato dextrose agar (PDA) in Petri dishes at 30 °C for 3 days, and mycelial disks (5 mm × 5 mm) were then used to inoculate in the seed medium (potato dextrose broth, PDB). Seed cultures were incubated in 250 mL Erlenmeyer flasks containing 100 mL of liquid medium. The flasks were shaken at 150 rpm, 28 °C for 3 days, and then 10 mL of the seed mycelium was used to inoculate 100 mL of basal medium (20 g soluble starch, 10 g peptone, 5 g yeast powder, 1 g K_2_HPO_4_, 1 g NaCl, 1000 mL H_2_O, and pH 7.4) in a 250 mL Erlenmeyer flask and cultured on a rotary shaker at 150 rpm, 28 °C for 5 days.

### 2.2. Amylase Assay

Amylase was assayed using soluble starch as a substrate. Fractions of 20 µL containing the enzyme solution were added to 180 µL substrate solution (1% soluble starch in 50 mM Tris-HCl, pH 7.5), and the mixture was then incubated at 50 °C for 10 min. The reaction was stopped by the addition of dinitrosalicylic acid (DNS) solution and boiled for 5 min [27]. After cooling, ddH_2_O was added up to 2.5 mL, and amylase activity was determined using a spectrophotometer at OD_540_. Glucose (from 0.2 µM to 2 µM) was used to construct a standard curve. Protein concentration was determined with the method established by Bradford using bovine serum albumin (BSA) as a standard protein [28]. One unit of amylase activity is defined as the amount of enzyme releasing 1 µmol of glucose per minute from soluble starch at 50 °C and a pH value of 7.5.

### 2.3. Experimental Design and Statistical Analyses

#### 2.3.1. Preliminary Experiments: The Effects of Carbon, Nitrogen, and Other Factors on Production of α-Amylase by *A. apis*

To determine the influence of carbon and nitrogen sources on amylase production by *A. apis*, different carbon sources (L-arabinose, D-glucose, D-galactose, D-sucrose, D-fructose, D-lactose, solublestarch, D-maltose, dextrin, chitin, D-trehalose, D-mannitol, D-sorbitol, D-xylitol, and citric acid) and nitrogen sources (tryptone, peptone, yeastpowder, casein, glycine, yeast extract, urea, (NH_4_)_2_S_2_O_8_,KNO_3_, NaNO_3_,NH_4_NO_3_, (NH_4_)_2_HPO_4_, and (NH_4_)_2_SO_4_) were examined independently while maintaining other compositions of basal medium constant. Metal salts (including KCl, MgCl_2_, NaCl, ZnCl_2_, CuCl_2_, MnCl_2_, CaCl_2_, FeCl_2_, FeCl_3_, and BaCl_2_), vitamins (involving C, E, PP, B_1_, B_2_, B_4_, B_5_, B_6_, and folacin), and pH values (at 6.0, 6.5, 7.0, 7.5, 8.0, and 8.5) were also evaluated for their effects on amylase production.

#### 2.3.2. Plackett–Burman Design

Plackett–Burman design (PBD) is a mathematical modeling approach employed for the optimization of biological processes [22,29]. Six independent and two dummy variables were selected for screening in 12 trials (Appendix A and Table 1). Each individual variable was designed at two levels: high (+1) and low (−1) (Appendix A). Dummy variables were used to estimate errors in the experiment. The effect of each parameter on amylase production was identified by the following equation:*E* = (∑*M*_+_ − ∑*M*_−_)/*N*
where *E* is the effect of the tested variable; *M_+_* is the amylase activity from the trials where the variable was present at a high concentration; *M_−_* is the amylase activity from the experiment where the variable was present at a low concentration; and *N* is the total number of trials.

#### 2.3.3. The Steepest Ascent Experiment

The method of steepest ascent is a procedure for moving sequentially along the path of steepest ascent, i.e., along the path of the maximum increase in the response [30]. Based on the results of PBD experiments, the optimal level of scope of every selected variable was explored by the mean of the steepest ascent method. The path started from the center of the PBD model and continued until the response no longer increased. The experimental design details are depicted in Table 2. This steepest ascent experiment covered five steps, as shown by the numbers 1–5 following the plus sign in Table 2. The paths of maltose, CaCl_2_, and pH values examined in relation to amylase production began at 45 g/L,1.39 g/L, and 7.0, respectively, with a step (Δ) of 0.5 g/L, 0.05g/L, and −0.2, respectively (Table 2).

This steepest ascent experiment included five steps, as illustrated by the numbers 1–5 following the plus sign. Values for amylase activity are the means (SE) of three replicate flasks.

#### 2.3.4. Box–Behnken Design

Box–Behnken design (BBD) allows for modeling of the relationships between multiple variables, decreasing the number of experimental runs [31]. To describe the level of each selected factor and the interactions among them that influence amylase production, the BBD methodology was performed, in which 3 factors and 3 levels were included. The ranges and levels of the respective factors studied in this research are given in Appendix A. The following second-order polynomial coefficients were fitted to the following model:Y = b_0_ + b_1_X_1_ + b_2_X_2_ + b_3_X_3_ + b_11_X_1_^2^ + b_22_X_2_^2^ + b_33_X_3_^2^ + b_12_X_1_X_2_ + b_13_X_1_X_3_ + b_23_X_2_X_3_,
where Y is the calculated response; b_0_ is the average effect; b_1_, b_2_, and b_3_ are the linear coefficients; b_11_, b_22_, and b_33_ are the quadratic coefficients; b_12_, b_13_, and b_23_ are the interaction coefficients. The validity of the fit to the regression equation was determined by the calculation of the coefficient R^2^. Fisher’s F-test at the 5% level of significance and the determination coefficient R^2^ were used to test the model’s adequacy.

#### 2.3.5. Statistical Analysis

All the experiments were performed in triplicate. Experimental designs and the polynomial coefficients were generated and analyzed using STATISTICAsoftware 9.2 (StatSoft Inc., Tulsa, OK, USA). A statistical analysis of the model was employed to evaluate the analysis of variance (ANOVA).

### 2.4. Purification of A. apis Amylase

*Ascosphaera apis* was inoculated into the above-mentioned optimized culture medium and incubated in a rotary shaker (150 rpm) at 28 °C for 3 days. The culture broth was centrifuged at 12,000 rpm for 10 min in a refrigerated centrifuge. The resultant cell supernatants were then used for the purification of enzymes. Ammonium sulfate was added to the crude culture supernatant to 50% saturation at 4 °C. After standing overnight at 4 °C, the precipitate was recovered by centrifugation (12,000 rpm, 10 min, 4 °C) and dissolved in a minimum volume (~10 mL) of 50 mM Tris buffer (pH 7.4) followed by dialysis (1 L, 12 h) against the same buffer.

The ammonium sulfate fraction was applied to DEAE-Sepharose A-52 anion exchange columns (2.0 cm × 20 cm) previously equilibrated with 50 mM Tris buffer (pH 7.4). The column was eluted with a linear gradient of 0–1 M NaCl in the same buffer at a flow rate of 0.5 mL/min. Fractions were assayed for amylase activity, and active fractions were pooled and concentrated by ammonium sulfate precipitation (as above). The ammonium sulfate-precipitated DEAE-pool (~100 mL) was loaded onto a Sephadex G-100 gel filtration column (1.0 cm × 60 cm), equilibrated in 50 mM Tris buffer (pH 7.4). The column was eluted with the same buffer at a flow rate of 15.0 mL/h. Fractions with active enzyme activity were pooled and used for further analysis.

### 2.5. Electrophoresis and Molecular Weight Determination

The molecular mass of the *A. apis* amylase was analyzed by sodium dodecyl sulfate- polyacrylamide gel electrophoresis (SDS–PAGE) [32]. Samples (15–20 μL) were mixed with 250–400μL of 4% SDS, 2% β-mercaptoethanol, glycerol 20%, 0.01 mol/L Tris buffer (pH 6.8), and 0.02% bromophenol blue in a total volume of 5-10 μL and heated at 100 °C for 5 min before electrophoresis. After electrophoresis, the gel was stained with Coomassie Brilliant Blue R-250. Protein standards contained seven proteins of 105, 90, 79, 66, 55, 45, and 17 kDa.

### 2.6. Characterization of the A. apis Amylase

The effect of temperature on enzyme activity was studied by conducting the assay at various temperatures ranging from 30 to75 °C. In the test of thermal stability, the enzyme solutions were incubated at the desired test temperature (30–75 °C). After 30 min, 20 µL enzyme aliquots were withdrawn from the reaction mixture and added to 180 µL 1% soluble starch (prepared in 50 mM Tris-HCl, pH7.5) to determine the residual amylase activity after cooling. The influence of pH on amylase activity was determined in a range of 3.0–10.0 using 0.1 M acetate buffer, phosphate buffer, Tris-HCl buffer, and glycine–NaOH buffer, respectively. The effect of pH value on stability was investigated by pre-incubating the enzyme solution at various pH values (ranging from 3.0 to 10.0) at 50 °C for 30 min, and the remaining amylase activity was then measured using the standard assay method.

The effects of various metal ions and organic solvents on the enzyme activity were studied by pre-incubating the enzyme with the indicated compound in 50 mM Tris-HCl buffer solution (pH 7.5) at 50 °C for 10 min. Activity was then determined via the enzyme assay described above. Kinetic parameters of the purified enzyme were verified by determining enzyme activity using different concentrations of soluble starch (in 50 mM Tris-HCl buffer, pH 7.5). Kinetic constants were calculated using the Lineweaver–Burke plots.

### 2.7. Determination of Amylase-Type and Amylase Plate Assay

To determine the type of amylase, product formation was examined by thin-layer chromatography (TLC) using glucose (0.1 µL of 15 mg/mL) and maltose (0.1 µL of 15 mg/mL) as standards. The purified enzyme was allowed to hydrolyze the soluble starch substrate for 10 h and 20 h, after which aliquots of the reaction mixture (0.1 μL) were spotted onto the TLC plate (Silica gel 60 F_254_ TLC-plates Merck, Germany). TLC plates were developed using chloroform/ethylic acid/H_2_O (6:7:1; *v*/*v*/*v*) as the mobile phase in a presaturated chamber. After separation was finished and the plate was dried, products were visualized via spray with the color-developing agent (2% diphenylamine dissolved in acetone (*w*/*v*):2% aniline dissolved in acetone (*w*/*v*): phosphate = 5:5:1 (*v*/*v*/*v*). Plates were dried using an air blower until a product (blue spots) appeared. The concentrations of hydrolysis products were analyzed by ZORBAX carbohydrate analysis column chromatography (5 μm, 4.6 × 250 mm, Agilent Technologies, Inc. Santa Clara, California, USA). The injection volume of diluted and filtered hydrolysate was 10 μL and the column temperature was set to 30 °C. The mobile phase was acetonitrile/water (7:3, *v*:*v*) mixed solvent with a flow rate of 1.0 mL/min.

Amylase activity was examined by an agar (1.5%) plate assay supplemented with 1% starch. Sterile filter papers (0.5 cm diameter) were spotted with amylase-containing solutions (0.1–0.2 mL) and placed onto the plate. The plate was then incubatedat 30 °C for 24 h, with H_2_O used as a blank control. After incubation, the diameter of the hydrolysis zone was measured.

### 2.8. Bioreactor Production of Amylase

*Ascosphaera apis* was inoculated in the seed medium (PDB) and the seed culture (5%, *v*/*v*), which was then used to inoculate a small-scale 3.7-L bioreactor (KLF2000, Bioengineering, Zürich, Switzerland) with a broth volume of 1.8 L containing the optimized media components. Cultivation was performed at 28 °C with an aeration rate of 1.0 vvm and an agitation speed of 140 rpm. Every 8 h, 1 mL of the culture solution was withdrawn from the bioreactor and used to measure amylase activity and residual maltose.

## 3. Results

### 3.1. Preliminary Experiments: The Effects of Carbon, Nitrogen, and Other Factors on Production of α-Amylase by A. apis

Amylase production was highest with the addition of maltose, yielding an amylase-specific activity of 11.21 ± 0.15 U/mL. Regarding the nitrogen source, amylase production was highest when yeast powder was used as the nitrogen source, with an amylase activity of 11.17 ± 0.23 U/mL. The other major factors affecting amylase production were Ca^2+^, vitamin E, and pH 7.5. Thus, the above variables were used to further study the optimization of the media.

### 3.2. Plackett–Burman Design

A set of preliminary experiments, as detailed in the Section 2, indicated five media composition variables that displayed significant effects on amylase production by *A. apis*. These components included the concentrations of carbohydrate growth source (maltose), yeast extract, CaCl_2_, vitamin E, and the pH of the media. A two-level PBD optimization modeling of these factors is given in Appendix A and Table 1, which yielded specific enzyme activities that ranged from 24.97 ± 1.04 to 37.92 ± 0.83 U/mL. Amlyase-specific activity increased as a function of maltose concentration but was inversely correlated with yeast extract concentration (Appendix A and Table 1).

The PBD modeling with the Pareto chart revealed maltose (X_1_), CaCl_2_ (X_4_), and pH (X_8_) as the key factors affecting amylase production (Table 1 and Appendix A), and the model fit showed a determination coefficient (*R*^2^) of 0.9789, indicating a robust prediction. To further define the optimal range of each critical parameter inside the present design space, a path of steepest ascent test was conducted. Levels of medium factors were varied in a series of five steps (1–5Δ). Amylase activity was highest (47 ± 1.6 U/mL) in the second step (X + 2Δ), when maltose and CaCl_2_ were increased and the medium pH was decreased (Table 2). These data were used to further optimize media composition (Appendix A).

### 3.3. Optimization of Amylase Production by Response Surface Methodology

The experimental findings resulting from the BBD protocols given in Appendix A were further analyzed (Appendix A), with accompanying ANOVA results (Appendix A). The results show that the model equation adequately depicts the response surface. The polynomial model for amylase production was mathematically expressed as follows:Y = 45.98438 − 0.965278X_1_ + 1.542966X_2_ − 0.092741X_3_ − 4.542247X_1_^2^ + 3.612153X_1_X_2_ + 2.790317X_1_X_3_ − 3.206559X_2_^2^ − 3.770565X_2_X_3_ − 4.440869X_3_^2^
where Y represents the response factor (amylase activity), and X_1_, X_2_, and X_3_ are the coded factors. ANOVA analysis (Appendix A) was adopted to describe the accuracy of the model terms and the model equation of the response surface. The Fisher’s *F*-test with a low probability value (< 0.05, as shown in Appendix A) denoted that the model was statistically significant. From the results of the ANOVA analyses, the model displayed a high determination coefficient (*R*^2^) of 0.9528. The value of the adjusted determination coefficient *R*^2^*_adj_*(0.868) in Appendix A also supported a high significance for the model.

The optimum conditions for all factors were achieved by employing a response surface model. Three-dimensional response surface graphs and the isoresponse curves for parameter pairs: maltose and CaCl_2_ (Figure 1A), maltose and pH (Figure 1B), and CaCl_2_ and pH (Figure 1C) were produced by plotting the response variation of each of the two coded variables, keeping the third variable constant at the optimal level.

Based upon the modeling and experimental verification, the optimum values for the three variables most strongly influencing amylase production were determined to be 45.99 g/L for maltose, 1.51 g/L for CaCl_2_, and a culture pH = 6.6. These conditions resulted in an experimentally calculated amylase activity of 46.25 ± 0.49 U/mL as compared to 4.2 U/mL for the original media, resulting in an overall ~11-fold increase in enzyme production.

### 3.4. Enzyme Purification and Characterization

Using the identified optimized media composition, a 0.5 L flask culture of *A. apis* was grown and used to purify the secreted amylase. The enzyme was purified approximately 14.5-fold from the cell-free culture supernatant in three steps as detailed in the Section 2, resulting in 1.61 mg of purified protein with an overall yield of 2.5%. Briefly, proteins in the crude extract were precipitated by ammonium sulfate (50%), followed by fractionation using DEAE-Sepharose A-52 and Sephadex G-100 gel filtration chromatography. A summary of purification steps, total activity, total protein, specific activity, purification fold, and % yield for each step is given in Table 3. The purity of the final amylase fraction was examined by SDS–PAGE (Figure 2A). The purified amylase had a molecular weight of ~75 kDa.

### 3.5. Characterization of the Purified Amylase

The purified *A. apis* amylase displayed maximum activity at pH 7.5 (Figure 2B), with a rather sharp temperature optimum of 55 °C at pH 7.5 (Figure 2C), with activity decreasing by 40–45% at 60 °C < T < 50 °C. The enzyme maintained > 80% of its original activity in a pH range of 7.0–10.0 after incubation at 4 °C for 24 h, but was progressively sensitive to lower pH conditions, ultimately having less than 10% residual activity below pH 4 (Figure 2B). The enzyme retained full activity when incubated (30 min) at temperatures ≤ 55 °C but displayed a sharp decrease at temperatures > 55 °C, with an 80% reduction in residual activity when incubated at temperatures > 65 °C for 30 min before being assayed (Figure 2C).

The effect of various metal ions on the enzyme activity of the purified amylase was examined in reaction mixtures containing 10 mM of the indicated compounds (Table 4). The addition of ferrous chloride to the reaction mixture increased the activity of the purified amylase by ~45%, whereas ferric chloride resulted in ~20% decrease in activity as compared to the unamended assay conditions. K^+^ had only a slight inhibitory effect, whereas Na^+^ decreased activity by ~20%. The addition of Mg^2+^ or Ca^2+^ inhibited activity by ~60%, whereas the addition of Zn^2+^ resulted in 95% inhibition, and the addition of Cu^2+^, Mn^2+^, or Ba^2+^ completely inhibited the enzyme. The enzyme was stable in a variety of organic solvents (tested at 10% final concentration; Table 3). Greater than 90% of the amylase activity was retained in solutions containing DMSO, isopropanol, dichloromethane, acetone, or methanol. Slight inhibition (15–25%) was seen in the presence of n-butyl alcohol, n-hexane, or ethanol, with ~35% inhibition seen in the presence of chloroform.

The kinetics of the purified amylase was investigated using different concentrations of soluble starch as a substrate. Under the optimal enzyme reaction conditions, *K*_m_ and *V*_max_ values of the purified *A. apis* amylase were calculated from Lineweaver–Burke double reciprocal plots to be6.22 mmol/L and 4.21 μmol/mL·min, respectively.

In order to determine the catalytic mechanism of the enzyme (α- or β-), the starch substrate hydrolysis products were analyzed by TLC, as detailed in the Section 2. The major products of the reaction were glucose and maltose. The concentration of carbohydrate in the hydrolysis product was determined by HPLC, followed by146.54 ± 12.39 mg/mL glucose and 189.73 ± 21.68 mg/mL maltose. It indicated the purified amylase was an endo-amylase (α-amylase) that randomly cleaved the starch substrate (Figure 3A). Enzyme activity was further confirmed using a plate assay in which the active fractions resulted in zones of clearing on starch agar plates (Figure 3B).

### 3.6. Bioreactor Production of the A. apis Amylase

Fermentation of the maltose growth substrate by *A. apis* and concomitant amylase production by the fungus were carried out in a 3.7-L bioreactor over a 48 h time period. Amylase activity and residual maltose were measured every eight hours. Culture maltose concentration slightly decreased from 8 h to 24 h but was more rapidly utilized after 24 h (Figure 4), suggesting an adaptation phase. Amylase activity increased from 8 h to 16 h, remained level from 16 h to 24 h, and then decreased from 24 h to 32 h, and remained level after 32 h (Figure 4). Enzyme production was up to 99.38 U/mL, which was almost 2-fold higher than that produced in flask cultures (46.25 U/mL). During the fermentation process, the culture pH did show a change (Figure 4).

## 4. Discussion

Amylases have been extensively sought after for use in a wide range of clinical and pharmaceutical applications, as well as in the food, detergent, textile, brewing, and distilling industries, because of their nontoxic characteristics [14,16,17]. Amylases have been characterized from almost every group of organisms, including from a wide range of eukaryotes, e.g., animals (including humans), plants, protists, and fungi, as well as from any prokaryotes and Archaea [33,34,35,36]. Fungi, in particular, have been rich sources of amylases with various desirable properties, e.g., thermal, pH, and salt stability, and have been widely used to meet current commercial needs [35,37]. The insect pathogenic fungus *A. apis* is an important and potentially dangerous pest of bees that has been plagued with population declines in various parts of the world. *A. apis* produces multiple hydrolases, including proteases, glycosidases, and esterases, but there are few reports of these enzymes (particularly α-amylases) that may play important roles in disease as well as potentially being useful in biotechnological applications [6,7,9]. Hence, to our best knowledge, this is the first report of the characterization of an α-amylase from this important fungus.

Optimization with respect to conditions that affect enzyme (in this case, amylase) production, including culture ingredients and conditions, is critical for increasing enzyme yields while decreasing production costs, but can be a time-consuming and costly process [37,38]. Response surface methodology (RSM) is a well-known tool that can be used to rapidly optimize medium components and other vital parameters responsible for enzyme production based on a set of mathematical and statistical algorithms [25]. In the present study, RSM was used with PBD to assess the relative importance of different medium ingredients for amylase production using a submerged culture of *A. apis*, and the crucial factors were determined as maltose, CaCl_2_, and pH value. Application of RSM is a facile approach to screening and optimizing multiple sets of variables [24,26]. From the results of the ANOVA analyses, the model displayed a high determination coefficient (*R*^2^) of 0.9528 (for the best-fit statistical model, the *R*^2^ value should approach 1.0 [22,39]). Furthermore, amylase activity (45.28 U/mL) was significantly greater (about 11-fold) with media comprising 46 g/L maltose, 1.51 g/L CaCl_2_, and pH 6.6. The above results indicated that it was of great application value for mass production of natural amylase using a submerged culture of *A. apis*.

The molecular weight of amylases from numerous fungi was previously reported to be in the range of 42–70 kDa [38]. The *A. apis* α-amylase appears to be slightly larger, with an approximate weight of 75 kDa. Moreover, the specific activity of the amylase was thermostable and in abroad pH range. The enzyme activity reached its maximum at 55 °C, equal to the level of thermophilic *Anoxybacillus flavithermus* [40]. Interestingly, the optimal temperature is similar to that of α-amylase purified from honey [41]. This similarity may be related to the environment where bees grow. Moreover, this enzyme was relatively insensitive to Fe^2+^ compared with other amylases [42], which still maintained 100% activity at a concentration of 10 mM Fe^2+^. The tolerance to Fe^2+^ and K^+^ of this α-amylase could even reach the level of halophilic α-amylase from *Klebsiella* sp. [43]. The *K_m_* value (6.22 mmol/L) of the enzyme located in the middle of 1.02–34.06 mmol/L was recorded for many fungal amylases [16,17,38]. The *K_m_* and *V_max_* values of the enriched α-amylase using soluble starch as the substrate here demonstrated that it possessed a moderately high affinity with soluble starch and reached *V_max_* at a relatively low substrate concentration. Notably, the amylase from *A. apis* exhibited higher in vitro amylolytic activity on starch agar plates, which makes it potentially versatile for use in different industrial operations.

In successive scale-up fermentation steps, α-amylase production was improved about 2.15-fold in comparison with that obtained in shake flasks, with an overall increased yield of >22-fold from the original media composition and culture conditions. These data indicate a very simple approach that can facilitate more rapid amylase development and production. The bioreactor system is novel and can solve some major problems associated with the solid-state fermentation (SSF) process. *A. apis* has been reported to date as the new fungus resource for α-amylase production.

## 5. Conclusions

Amylases are critical industrial enzymes used in a variety of biotechnological applications. Hydrolases have also been implicated as virulence factors in a number of fungi, either for penetrating host structures or for assimilating host nutrients, and since *A. apis* is an important fungal pathogen of bee brood that can result in significant colony losses, particularly in connection with other diseases and colony stress, the characterization of these enzymes may provide new avenues for controlling the fungal disease. Here we provide a rational design basis for optimizing the production of a secreted enzyme (amylase) from a fungal source. The combination of RSM and Box–Behnken design was used to reduce the experimental conditions needed to be tested in order to maximize enzyme production. Based upon a small series of preliminary experiments that were used to feed into the modeling designs, several critical factors were identified and further experimentally examined. The final enzyme yield was > 10-fold higher than the non-optimized media conditions.

The enzyme was purified to homogeneity and demonstrated to act as an α-amylase with a mid-range temperature optimum (~55 °C), is stable at high pH (> 7.5), and is sensitive to low pH (< 6.5). Fe^2+^ stimulated enzyme activity, but the enzyme was sensitive to Zn^2+^, Cu^2+^, Mn^2+^, and Ba^2+^. The enzyme was also stable in a wide range of organic solvents (up to the 10% tested). These results suggest that the α-amylase from *A. apis* has potential for application in the textile, food, and distilling industries. Future studies probing the role of the enzyme in *A. apis* physiology and development as well as exploring applications of the enzyme are warranted.

## Figures and Tables

**Figure 1 jof-09-01082-f001:**
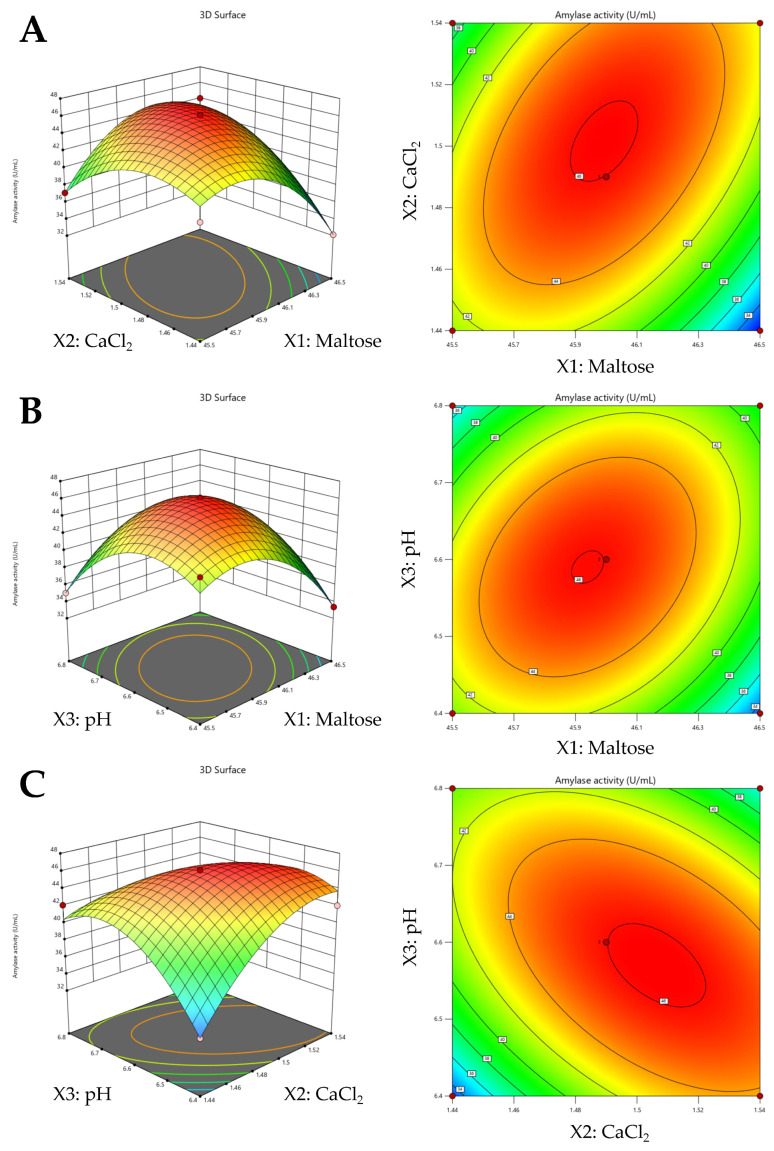
(**A**) Interactive effects of maltose (X_1_) and CaCl_2_(X_2_) on *A. apis* amylase activity (as predicted by 3-D response surface and 2-D contour plots).(**B**) Interactive effects of maltose (X_1_) and pH(X_3_) on *A. apis* amylase activity (as predicted by 3-D response surface and 2-D contour plots).(**C**) Interactive effects of CaCl_2_ (X_2_) and pH (X_3_) on *A. apis* amylase activity (as predicted by 3-D response surface and 2-D contour plots).

**Figure 2 jof-09-01082-f002:**
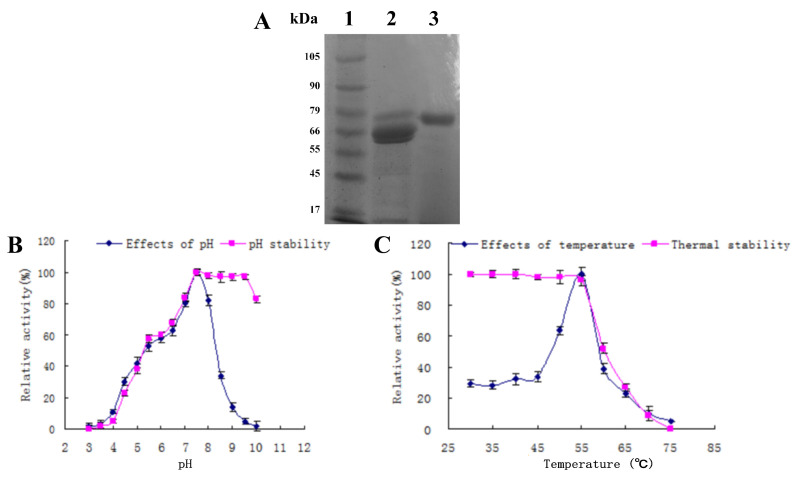
(**A**) Results of SDS-PAGE of purified amylase from *A. apis*. Lane 1: molecular weight markers. Lane 2: crude amylase. Lane 3: purified amylase. (**B**) Effects of pH on the activity and pH stability of purified amylase from *A. apis*. Each value in the graph represents the mean ± SD. All experiments were performed in triplicate. (**C**) Effects of temperature on the activity and thermal stability of purified amylase from *A. apis*. Each value in the graph represents the mean ± SD. All experiments were performed in triplicate.

**Figure 3 jof-09-01082-f003:**
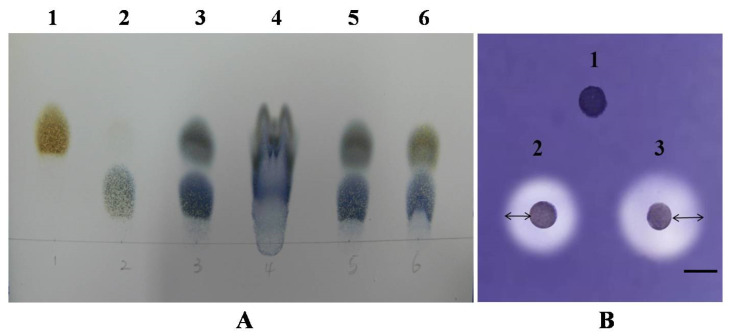
(**A**) Thin-layer chromatography (TLC) analysis of the hydrolysis of soluble starch by purified amylase from *A. apis*. The standards used were glucose (Lane 1) and maltose (Lane 2); products of hydrolysis of soluble starch by α-amylase standard (Lane 3) and β-amylase standard (Lane 4); Lane 5: products of hydrolysis of soluble starch by purified *A. apis* amylase incubated for 20 h; Lane 6: reaction products after 10 h incubation. (**B**) Representative image of the amylolytic activity plate assay; 200 μL was spotted onto starch agar plates and the plate was incubated at 30 °C for 24 h. 1—control; 2—purified amylase; 3—purified amylase. The arrow indicates the zone of clearance.

**Figure 4 jof-09-01082-f004:**
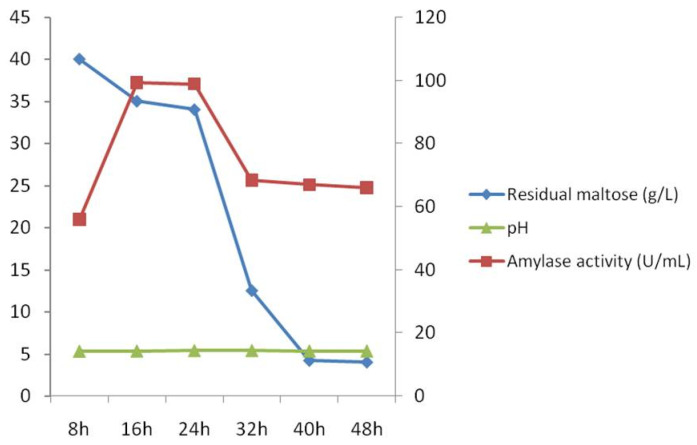
Time profile of growth of *Ascosphaera apis*, residual maltose, pH values, and amylase activity in bioreactor cultivation using optimized medium at an aeration rate of 1.0 vvm and an agitation speed of 140 rpm.

**Table 1 jof-09-01082-t001:** *A. apis* amylase activity as affected by levels of medium variables and as predicted by Plackett–Burman design.

Code	Medium Variable	Levels (g/L)		Statistics for Observed Data	
		Low (−1)	High (+1)	*t*	Pr > |*t*|
X_1_	Maltose	40	50	3.407537	0.0422 *
X_2_	Yeast powder	20	30	−2.84837	0.0652
X_3_	Dummy	−1	1	−1.13515	0.3388
X_4_	CaCl_2_	1.11	1.67	3.799317	0.0320*
X_5_	ZnCl_2_	4.93	7.39	−0.78043	0.4921
X_6_	Dummy	−1	1	−1.1057	0.3496
X_7_	V_E_	0.05	0.075	−2.78186	0.0689
X_8_	pH	7.0	7.5	−4.78119	0.0174 *

* Statistical significance for comparison of low vs. high levels at the 95% probability level.

**Table 2 jof-09-01082-t002:** Design and results of the steepest ascent test for *A. apis* amylase activity.

Step Change Value	Maltose (gL^−1^)	CaCl_2_ (gL^−1^)	pH	Amylase Activity (U/mL)
X	45.0	1.39	7.0	41.38 (1.62)
△	0.5	0.05	−0.2	0
X + 1△	45.5	1.44	6.8	44.18 (1.39)
X + 2△	46.0	1.49	6.6	46.68 (1.55)
X + 3△	46.5	1.54	6.4	44.61 (1.07)
X + 4△	47.0	1.59	6.2	43.65 (1.91)
X + 5△	47.5	1.64	6.0	41.38 (2.26)

**Table 3 jof-09-01082-t003:** Purification and activity of amylase from *A. apis*.

Fraction	Total Activity (U)	Total Protein (mg)	Specific Activity (U/mg)	Purification (Fold)	Yield (%)
Crude enzyme	2082.62	80.68	25.81	1	100
50% ammonium sulfate	1008.77	9.97	101.18	3.92	24.96
Ion exchange chromatography	692.05	2.14	323.39	12.53	7.12
Gel filtration chromatography	597.55	1.61	371.15	14.38	2.51

**Table 4 jof-09-01082-t004:** Effect of metal ions and organic solvents on the activity of purified amylase from *A. apis*.

Metal Ion (10 mM)	Relative Activity (%)	Organic Solvent (10%)	Relative Activity (%)
None	100 (0.14)	none	100 (0.11)
Fe^2+^	143.25 (2.12)	DMSO	99.14 (0.82)
K^+^	91.73 (1.96)	isopropanol	96.98 (1.21)
Fe^3+^	81.91 (0.73)	dichloromethane	95.31 (0.44)
Na^+^	77.46 (2.21)	acetone	91.97 (1.15)
Mg^2+^	37.99 (0.65)	methanol	91.49 (1.53)
Ca^2+^	37.99 (0.87)	n-butyl alcohol	86.26 (0.77)
Zn^2+^	5.74 (0.38)	n-hexane	83.67 (1.49)
Cu^2+^	0.00	ethanol	75.70 (0.68)
Mn^2+^	0.00	chloroform	65.63 (0.93)
Ba^2+^	0.00	-	-

## Data Availability

Not applicable.

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
