# Peer review of "Characterization of an α-Amylase from the Honeybee Chalk Brood Pathogen Ascosphaera apis"

_jof, 2023, doi:10.3390/jof9111082_

Round 1

Reviewer 1 Report

Comments and Suggestions for Authors

This manuscript describes biochemical characterization of the enzyme alpha-amylase from the fungus Ascosphaera apis, which is the causal agent of the bee chalkbrood disease. The fungus utilizes hydrolytic enzymes to digest substances from the host. A. apis alpha-amylase was produced in optimized conditions, the optimization involved a rational design strategy based on preliminary experiments with different carbon and nitrogen sources, metal ions and vitamins followed by mathematical modeling (assessing the relative importance of different medium ingredients) to achieve the highest activity yield. As a reviewer, I am not fully oriented in the described calculating approaches but they seem adequate to me. The reaction products were analyzed by thin layer chromatography.

There are a few comments from my side to be considered for a revision:

Check for spelling errors

Figure 2: What are the corresponding purification steps providing the samples separated in the lanes 2 and 3 (denoted as “crude amylase”)? Please specify.

I do not see error bars in Figs. 2B and 2C.

Table 3: Why lower concentrations of metal ions were not tested such as 100 μM or 1 mM for those ions, which completely inhibited at the level of 10 mM?

Page 11, bottom sentence: I would like to see the Km a Vmax values provided in standard units, such as mmol/L and μmol/min, respectively.

In the discussion section the authors compare their results with previous data on amylases only as regards to the molecular mass of the enzyme. I would like to see an additional discussion particularly with respect to the kinetic data such as the determined pH and temperature optima or sensitivity to inhibitors.

Comments on the Quality of English Language

There are no additional comments.

Author Response

Dear Editors and Reviewers:

Thank you for your letter and comments relating to our manuscript entitled “Characterization of an α-amylase from the honey bee chalk brood pathogen Ascosphaera apis” (ID: jof-2687324). The comments were very valuable in addition to being helpful for revising and improving our manuscript, as well as the important guiding significance to our research. We have read the comments carefully and made corrections accordingly. Revised parts are marked in blue in the manuscript. The main corrections in the paper and our responses to the reviewer’s comments are given below. We hope that the revisions in the manuscript and our accompanying responses will be sufficient to make our manuscript suitable for publication in Journal of Fungi.

Responses to the comments of the reviewers:

Reviewer 1

Comments 1: Figure 2: What are the corresponding purification steps providing the samples separated in the lanes 2 and 3 (denoted as “crude amylase”)? Please specify.

Response 1: We have changed it as shown in the following figure.

Figure 2 A. Results of SDS-PAGE of purified amylase from A. apis. Lane 1: molecular weight markers. Lane 2: crude amylase. Lane 3: purified amylase.

Comments 2: I do not see error bars in Figs. 2B and 2C.

Response 2: We have added error bars in Figs. 2B and 2C.

Comments 3: Table 3: Why lower concentrations of metal ions were not tested such as 100 μM or 1 mM for those ions, which completely inhibited at the level of 10 mM?

Response 3: As shown in Table 4, different metal ions demonstrated different degrees of inhibitory influences on enzyme activity at the concentration of 10 mM. However, different relative activities (21.31%, 33.48%, 37.65%) also could be respectively tested with the treatment of Cu2+, Mn2+ and Ba2+ at the level of 1 mM.

Comments 4: Page 11, bottom sentence: I would like to see the Km a Vmax values provided in standard units, such as mmol/L and μmol/min, respectively.

Response 4: We have changed the sentence.

Under the optimal enzyme reaction conditions, Km and Vmax values of the purified A. apis amylase were calculated from Lineweaver–Burke double reciprocal plots to be 6.22 mmol/L and 4.21 μmol/mL·min, respectively.

Comments 5: In the discussion section the authors compare their results with previous data on amylases only as regards to the molecular mass of the enzyme. I would like to see an additional discussion particularly with respect to the kinetic data such as the determined pH and temperature optima or sensitivity to inhibitors.

Response 5: We have added some kinetic data and related references to the discussion section.

The molecular weight of amylases from numerous fungi is previously reported with the range of 42-70 kDa [38]. The A. apis amylase appears to be slightly larger with an approximate weight of 75 kDa. Moreover, the specific activity of the amylase was thermostable and in the broad pH range. The enzyme activity reached its maximum at 55 oC, equal to the level of thermophilic Anoxybacillus flavithermus [40]. This enzyme was relatively insensitive to Fe2+ compared with other amylases [41], which still maintained 100% activity at the concentration of 10 mM Fe2+. The Km value (6.22 mmol/L) of the enzyme located in the middle of 1.02-34.06 mmol/L recorded for many fungal amylases [16, 17, 38]. The Km and Vmax values of the enriched α-amylase using soluble starch as the substrate here demonstrated that it possessed a moderately high affinity with soluble starch and reached Vmax at a relatively low substrate concentration. Notably, the amylase from A. apis exhibited higher in vitro amylolytic activity on starch agar plates, which makes it potentially versatile for use in different industrial operations available.

  1. Nam, Y.; Barnebey, A.; Kim, H.K.; Yannone, S.M.; Flint, S. Novel hyperthermoacidic archaeal enzymes for removal of thermophilic biofilms from stainless steel. J. Appl. Microbiol. 2023. 10.1093/jambio/lxad106

41.Kim, J.K.; Lee, C.; Lim, S.W.; Adhikari, A.; Andring, J.T.; McKenna, R.; Ghim, C.M.; Kim, C.U. Elucidating the role of metal ions in carbonic anhydrase catalysis. Nat. Commun. 2020, 11, 4557. 10.1038/s41467-020-18425-5

We tried our best to improve the manuscript and made some changes marked in blue in revised paper which will not influence the content and framework of the paper. We appreciate for Editors/Reviewers’ warm work earnestly and hope the revision will meet with your approval. Once again, thank you very much for your comments and suggestions.

Kind regards,

Junzhi Qiu

E-mail address: [email protected]

Reviewer 2 Report

Comments and Suggestions for Authors

The manuscript "Characterization of an α-amylase from the honey bee chalk brood pathogen Ascosphaera apis" is well written and structured. However, some modifications are requested:

1) The penultimate sentence of the Introduction: "Amylase production in A. apis..." is a Result and not an Introduction. Remove it;

2) Add space between the temperature and the unit of measurement throughout the manuscript;

3) Material and methods, item 2.3.1

The last paragraph: "Amylase production was highest..." is Result and not Material and methods;

4) Place the Tables and Figures in ascending order. Example: Table 1, 2, 3... and not as Table 1A, 1B...

5) Add Pareto charts;

6) Fig. 1A: change the color scale to allow visualize the result;

7) Fig. 2A B and C: there is no standard deviation in the images;

8) Fig. 3A: I recommend that carbohydrate analysis be performed by HPLC to quantify the results.

Author Response

Dear Editors and Reviewers:

Thank you for your letter and comments relating to our manuscript entitled “Characterization of an α-amylase from the honey bee chalk brood pathogen Ascosphaera apis” (ID: jof-2687324). The comments were very valuable in addition to being helpful for revising and improving our manuscript, as well as the important guiding significance to our research. We have read the comments carefully and made corrections accordingly. Revised parts are marked in blue in the manuscript. The main corrections in the paper and our responses to the reviewer’s comments are given below. We hope that the revisions in the manuscript and our accompanying responses will be sufficient to make our manuscript suitable for publication in Journal of Fungi.

Responses to the comments of the reviewers:

Reviewer 2

Comments 1: The penultimate sentence of the Introduction: "Amylase production in A. apis..." is a Result and not an Introduction. Remove it;

Response 1: We have removed the sentence.

Comments 2: Add space between the temperature and the unit of measurement throughout the manuscript;

Response 2: We have added space.

Comments 3: Material and methods, item 2.3.1

The last paragraph: "Amylase production was highest..." is Result and not Material and methods;

Response 3: We have moved the paragraph to Results section.

Comments 4: Place the Tables and Figures in ascending order. Example: Table 1, 2, 3... and not as Table 1A, 1B...

Response 4: We have changed them in ascending order.

Comments 5: Add Pareto charts;

Response 5: We have added Pareto charts.

Comments 6: Fig. 1A: change the color scale to allow visualize the result;

Response 6: We have changed the color scale of Fig.1A.

Comments 7: Fig. 2A  B and C: there is no standard deviation in the images;

Response 7: We have added the standard deviation in Fig.2 B and C.

Comments 8: Fig. 3A: I recommend that carbohydrate analysis be performed by HPLC to quantify the results.

Response 8: We have added the results of carbohydrate analysis performed by HPLC.

The concentration of carbohydrate in the hydrolysis product was determined by HPLC (Agilent 1260, USA), followed with 146.54 ± 12.39 mg/mL glucose and 189.73 ± 21.68 mg/mL maltose.

We tried our best to improve the manuscript and made some changes marked in blue in revised paper which will not influence the content and framework of the paper. We appreciate for Editors/Reviewers’ warm work earnestly and hope the revision will meet with your approval. Once again, thank you very much for your comments and suggestions.

Kind regards,

Junzhi Qiu

E-mail address: [email protected]

Round 2

Reviewer 2 Report

Comments and Suggestions for Authors

The manuscript "Characterization of an α-amylase from the honey bee chalk brood pathogen Ascosphaera apis" has undergone partial changes. Not all recommendations were met:

1) Although the authors mention that they added the Pareto graphs, they are not in the manuscript.

2) Replace with colors that make the result clear in Fig. 1.

4) Add the HPLC methodology in Material and Methods;

5) Add a space in "ion exchange" in Table 3.

Author Response

Dear Editors and Reviewers:

Thank you for your letter and comments relating to our manuscript entitled “Characterization of an α-amylase from the honey bee chalk brood pathogen Ascosphaera apis” (ID: jof-2687324). The comments were very valuable in addition to being helpful for revising and improving our manuscript, as well as the important guiding significance to our research. We have read the comments carefully and made corrections accordingly. Revised parts are marked in blue in the manuscript. The main corrections in the paper and our responses to the reviewer’s comments are given below. We hope that the revisions in the manuscript and our accompanying responses will be sufficient to make our manuscript suitable for publication in Journal of Fungi.

Responses to the comments of the reviewers:

Reviewer 2 (Round 2)

Comments 1: Although the authors mention that they added the Pareto graphs, they are not in the manuscript.

Response1: We have added the Pareto chart in the supplementary material and mentioned it in the revised manuscript.

The PBD modeling with Pareto chart revealed maltose (X1), CaCl2 (X4), and pH (X8) as the key factors affecting amylase production (Table 1 and Figure S1).

Figure S1. Pareto chart of the standardized effects for amylase activity.

Comments 2: Replace with colors that make the result clear in Fig. 1.

Response 2: We have changed the color scale of Fig.1.

Comments 3: Add the HPLC methodology in Material and Methods.

Response 3: We have added the HPLC methodology in Material and Methods.

The concentrations of hydrolysis products were analyzed by ZORBAX carbohydrate analysis column chromatography (5 μm, 4.6 × 250 mm, Agilent, USA). The injection volume of diluted and filtered hydrolysate was 10 μL and the column temperature was set to 30 oC. The mobile phase was acetonitrile/water (7:3, v:v) mixed solvent with a flow rate of 1.0 mL/min.

Comments 4: Add a space in "ion exchange" in Table 3.

Response4: We have added the space.

Table 3. Purification and activity of amylase from A. apis.

Fraction

Total

activity(U)

Total

protein(mg)

Specific

activity(U/mg)

Purification

(fold)

Yield (%)

Crude enzyme

2082.62

80.68

25.81

1

100

50% ammonium

 sulfate

1008.77

9.97

101.18

3.92

24.96

Ion exchange

chromatography

692.05

2.14

323.39

12.53

7.12

Gel filtration

chromatography

597.55

1.61

371.15

14.38

2.51

We tried our best to improve the manuscript and made some changes marked in blue in revised paper which will not influence the content and framework of the paper. We appreciate for Editors/Reviewers’ warm work earnestly and hope the revision will meet with your approval. Once again, thank you very much for your comments and suggestions.

Kind regards,

Junzhi Qiu

E-mail address: [email protected], and [email protected]
